# Multiple Aspects of the Fight against the Red Palm Weevil in an Urban Area: Study Case, San Benedetto del Tronto (Central Italy)

**DOI:** 10.3390/insects14060502

**Published:** 2023-05-30

**Authors:** Luca Bracchetti, Paolo Cocci, Francesco Alessandro Palermo

**Affiliations:** 1Unità di Ricerca e Didattica San Benedetto del Tronto (URDiS), University of Camerino, Via A. Scipioni, 6, 63034 San Benedetto del Tronto, Italy; 2School of Biosciences and Veterinary Medicine, University of Camerino, Via Gentile III da Varano, 62032 Camerino, Italy; paolo.cocci@unicam.it (P.C.); francesco.palermo@unicam.it (F.A.P.)

**Keywords:** red palm weevil, pesticides, potential chemical risk, urban environment, GIS, spatial distribution analysis

## Abstract

**Simple Summary:**

The red palm weevil beetle (*Rhynchophorus ferrugineus*), native to Asia, has assumed highly invasive behavior in the Mediterranean Sea basin, causing great damage to both date crops and ornamental plants. Given the lack of natural antagonists here, chemical substances must be administered to prevent attacks or cure infested palms. Through a multidisciplinary approach, we analyzed the evolution of the distribution of this beetle in San Benedetto del Tronto, a coastal city in central Italy rich in Canary Island date palms, considering both the control strategies used and their potentially negative effects. We observed that this beetle drastically reduced the palm stock by about half over the course of the 15 years between its arrival and the study’s conclusion in 2020. The local municipality’s program of chemical treatments has been very effective against new palm tree attacks but is expensive and poses toxicity risks for insects, animals, and the environment. However, currently, these treatments are the only effective tool for preserving the cultural landscape of this area. The challenge is to find the right balance between all these aspects.

**Abstract:**

The fight against alien invasive insect pests of plants in the urban environment often affects varied sectors of the economy, landscape gardening, public health, and ecology. This paper focuses on the evolution of the red palm weevil in San Benedetto del Tronto, a coastal urban area in central Italy. We investigated the evolution of this insect pest of palm trees in the 2013–2020 period, considering both the effectiveness of the chemicals used and their potentially harmful effects. With a multidisciplinary approach, we carried out a spatio-temporal analysis of the extent and mode of pest spread over time using historical aerial photos, freely available remote sensing images, and field surveys integrated in a GIS environment. We also assessed the toxicity risk associated with the chemicals used to protect the palms from the red weevil. The fight against this weevil is now concentrated in specific areas such as parks, roads, villas, hotels, farmhouses, and nurseries. The preventive chemical treatments applied are very effective in preserving the palms, but they show a toxic potential for all organisms. We discuss current local management of this pest, focusing on several aspects involved in the fight against this beetle in an urban area.

## 1. Introduction

The red palm weevil (RPW), *Rhynchophorus ferrugineus* Olivier, 1790 (Coleoptera: Curculionidae), is one of the most harmful palm pests, and its damage to date palms has been recorded since the mid-1980s in Saudi Arabia and then in Egypt in the 1990s [1]. Since 2000, the RPW has spread in the Mediterranean basin [2,3], infesting palms primarily belonging to the *Phoenix* genus that are used for ornamental purposes. The RPW is highly fecund and lays about 200 eggs on the top of the palm tree [2], and about 75% of these hatch [4], usually in 3 or 4 days [5]. There can be eggs from several females on a palm tree, from multiple generations [6]. The larvae are particularly harmful to the tree because they eat the soft tissues, especially the meristematic tissue, and in cases of severe infestation, they can kill the palm. The RPW tends to live from 1 to 3 months [7,8]. They can detect weak or infested palms and reach them through flights of over 900 m [9,10,11,12]; one study reported that they can cover overall flight lengths of more than 10 km [13].

The fight against the red palm weevil is unfortunately still an open question; early detection of the RPW is challenging because larvae are endophytic [14]. Several control methods have been used and tested so far, including biological control by means of entomopathogenic nematodes (e.g., Steinernematidae and Heterorhabditidae), fungi (e.g., *Beauveria bassiana*, *Metarhizium anisopliae*), viruses (*Baculovirus oryctes*, *Cytoplasmic Polyhedrosis Virus*) [15], bacteria (*Bacillus* sp., *Pseudomonas aeruginosa*), and mass trapping [16,17,18,19,20,21]. Nevertheless, to date, insecticide applications are the most effective method for protecting palms from palm weevil [2]. Insecticides can be applied outside the apical part of the plant, either with a high pressure (spray) or with a low pressure of a water-based insecticide solution [16,22,23,24,25,26,27] or inside it (trunk injection method) [28,29]. In the first case, treatment should be repeated periodically, depending on the local climate and the persistence of the sprayed insecticide [30]. With the trunk injection method, treatment may be more spaced out over time [31]. Endotherapy is regarded as a safer method than spray diffusion as it has fewer adverse effects on non-targeted species, humans, and the environment [31].

In the Marche region, many coastal resort towns have beautiful palm-lined seashore avenues, and this image is an important part of their appeal to tourists. For instance, the municipal area of San Benedetto del Tronto is known as the Palm Tree Coast, and city planners feel they must preserve this palm tree heritage at all costs.

In this region, chemical control of RPW started in 2008, using 15–20 L per palm of an insecticide spray solution containing chlorpyrifos (CPF) or acetamiprid (ACE)/imidacloprid (IMI) as the main chemical active substances. At least one annual application has been conducted during the last few years [32]. The widespread use of these synthetic pesticides in our study area has raised concerns over their toxic effects on the environment. CPF is one of the most widely used chlorinated organophosphate (OP) pesticides, and it is well known as a highly persistent chemical, especially in aquatic ecosystems [33,34]. The effects of CPF toxicity in both human and animal studies range from neurological dysfunctions to endocrine disruption [35]. ACE and IMI are two neonicotinoid insecticides that act as selective agonists of type-2 nicotinic acetylcholine receptors in insects [36]. However, the distribution of these compounds in the plant’s pollen and nectar [37] contributes to the exposure of other non-targeted invertebrates, such as pollinating honeybees (*Apis mellifera*) [38,39]. In addition, soil drench and seed treatment may contribute to spreading the toxins, thus also exposing vertebrate species, including humans [40]. 

Urban areas offer specific challenges for the management of pest infestations such as RPW. First, pesticide use is highly regulated to preserve public health, particularly in urban areas, and thus pest management with these substances may have to be limited in order to protect the citizenry. Second, infested palms are often on private land, which may complicate control programs [2] and research studies, as owners may not allow insecticide treatment on their property or collaborate with researchers.

In this regard, space or aerial surveys of palms offer a practical alternative. Sequential photo-surveillance from satellites [41] or drones [2] can be used to monitor the evolution of this pest and allow for the quantification of pest spread patterns and palm tree mortality rates [42,43]. In this paper, we present the evolution of the RPW spread (at 2020) in the San Benedetto del Tronto municipality (known as the Palm Tree Coast) with respect to the previous period we studied, 2007–2012 [41], using available remote sensing images, field surveys, and GIS. Moreover, we predict the potential adverse health effects that may result from exposure to chemicals used for RPW control by carrying out a system toxicology analysis. The specific objectives of this work were the following: a) to evaluate the effectiveness of the chemical treatments carried out by the municipality of San Benedetto del Tronto; b) to analyze the new RPW spatial distribution in this urban area; c) to predict potential adverse effects of the use of these insecticides, specifically regarding the signalling pathways they target; d) to suggest specific urban management measures. As far as we know, this was the first study to consider the multiple aspects of the battle against the RPW in an urban area.

## 2. Materials and Methods

### 2.1. The Study Area and Collected Data

The municipality of San Benedetto del Tronto, on the Adriatic coast of Central Italy (Figure 1), has a Mediterranean bioclimate (upper Meso-Mediterranean thermotype) [44] with annual rainfall and temperature averages of 551 mm and 15.8 °C, respectively, dry summers with average temperatures of 24 °C, and rainy and moderately cold winters with average temperatures of 7 °C [45].

At the end of 2020, all the *Phoenix canariensis* palm trees in San Benedetto del Tronto municipality were re-assessed and re-classified in terms of contagion (health, dead, infested), starting from the map generated from the July 2007 to July 2013 study of RPW distribution [41], through aerial photos of Google Earth time series, Google Street View time series, and subsequent field surveys.

Interviews with municipal employees and analysis of the municipal administration’s actions [46] served to define the measures taken against the RPW in 2013–2020 in this area. It was therefore possible to classify the palms in terms of treatment (treated or untreated) and to estimate the quantities and types of pesticides used.

### 2.2. Geo-Statistical Analysis

To analyze the recent distribution of the RPW in dead or infested palms with respect to the chemical treatments carried out (treated or untreated palms), the new collected data were entered into a previous GIS (ESRI^®^ ArcGIS© 10.1 [47]).

Hot spot analysis (Getis-Ord Gi*), expressed by Equation (1), was performed by the Hot Spot Analysis tool to highlight statistically significant areas of clustering (not infestation; chemical treatment), based on a new output shape file that reports a z-score, a *p*-value, and a confidence level of aggregation or dispersion bin field (Gi_Bin) for each feature.
(1)$$G_{i}^{*}=\frac{\sum_{j=1}^{n}w_{i,j}x_{j}-\overset{-}{X}\sum_{j-1}^{n}w_{i,j}}{S\sqrt{\frac{\left[n\sum_{j-1}^{n}w_{i,j}^{2}-\left(\sum_{j-1}^{n}w_{i,j}\right)\right.}{n-1}}}$$

*x_j_* is the attribute value for feature *j*; *w_i_,_j_* is the spatial weight between feature *i* and *j*; and *n* is equal to the total number of feature and:$$\overset{-}{X}=\frac{\sum_{j=1}^{n}x_{j}}{n}$$
$$S=\sqrt{\frac{\sum_{j=1}^{n}x_{j}^{2}}{n}}-{\left(\overset{-}{X}\right)}^{2}$$

The analysis returns the interpretation of these quantities in the Gi-bin values, expressed as a percentage of statistical confidence of aggregation or dispersion [high z-score and small *p*-value: spatial cluster with high values (hotspot); low negative z-score and small *p*-value: spatial clustering with low values (coldspot); z-score near zero: no apparent spatial clustering].

To consider a statistically significant hotspot, we considered a confidence level ≥95%, z-score ≥ 1.96 and a *p*-value < 0.05.

In our case, we performed two hot spot analyses, one for the healthy palms and another for the treated palms.

For this reason, the input shape file was modified by adding short numerical attribute fields based on the classification of all the palms in terms of the contagious event (1, healthy palm; 0, infested or dead palm) and chemical treatment (1, treated palm; 0, untreated palm).

On the basis of the processed data, the hot spot analysis was set up as follows: Conceptualization of Spatial Relationships: Inverse distance (nearby neighboring features have a larger influence on the computations for a target feature than features that are far away); Distance method: Euclidean distance; Standardization: none; Distance Band or Threshold Distance: none (when this parameter is left blank, the threshold value of the Euclidean distance that ensures every feature has at least one neighbor is automatically computed and applied).

Through overlapping of the resulting maps (the two generated shapefiles displayed by the Gi_Bin values), it was possible to verify whether a possible data cluster of healthy palms is superimposed on a possible data cluster of treated palms.

For this purpose, we generated a map formed of 20 m × 20 m units and used different colors to report information about “healthy and treated” and “healthy and not treated” palms; in this context, we also generated an index based on the percentage of overlap.

Finally, to support this evaluation, statistical analysis was performed for PRW infections on the palm trees. The risk factor affecting RPW occurrence was determined using the Chi-square test of independence between the status of each palm (infested or dead; uninfested) and a different treatment (chemical treatment yes or no). The Chi-square test was performed according to the required parameters: no more than 20% of the cells can have an expected frequency less than five, no cell can have an expected frequency less than one, and the sample size is sufficiently large [48,49]. In our study, the null hypothesis for this test stated that the occurrence of infection does not depend on the chemical treatments (treated vs. untreated).

### 2.3. In Silico Prediction of Insecticide Toxicity

IMI and CPF were investigated using an integrative system toxicological model to predict biological effects on human, bird, amphibian, fish, and insect systems. We used existing data from the Search Tool for Interactions of Chemicals database (STITCH v5.0) containing the interactions between chemicals and proteins to develop a network-based predictive model of the selected pesticides with targets. This approach has also been widely accepted to study chemical-gene interactions [50,51]. STITCH acts as a probabilistic network by collecting interactions from multiple sources, including experimental evidence, databases, and published data. The overall confidence score ranges from 0 to 1, where a value of 0.15 is considered as low confidence, 0.4 as medium confidence, and a score equal to or higher than 0.7 is regarded as high confidence. To obtain a reliable set of interactions, we removed all interactions with a confidence score lower than 0.7 and pesticide connections with uncharacterized protein. IMI and CPF were annotated with a canonical SMILES (Simplified Molecular Input Line Entry System) code retrieved from PubChem (C1CN(C(=N1)N[N+](=O)[O-])CC2=CN=C(C=C2)Cl; CCOP(=S)(OCC)OC1=NC(=C (C=C1Cl)Cl)Cl, respectively). In order to clarify the relationship between the selected pesticides and protein and their involvement in signalling pathways, we represented data using InterPro database, GO enrichment and KEGG pathway enrichment analyses provided by STITCH.

## 3. Results and Discussion

### 3.1. RPW Occurrence

Table 1 shows the classification of the palms according to their state (healthy/dead-infested) and the type of treatment (chemical treatment, yes or no). Since the beginning of the infestation (2007), more than half of the palms have died, and of these, just over 60% died after 2013. Since the beginning of the infestation in 2007 to date, more than half of the palms (3943 of 7385) have died, notwithstanding the chemical treatments that began to be administered in 2007, and of these, just over 60% (2456 of 3943) died after 2013.

Since the second half of 2013, the municipality of San Benedetto del Tronto has systematically applied chemical treatments four times a year to each palm along the “Dei Mille” seashore avenue (Table 2), in the city center and in the urban parks (1627 palms in total).

At the end of 2020, about 90% (89.73) of treated palms and about 35% (34.42) of untreated palms were not infested by the RPW. It should be noted that the latter percentage drops to 16.5% if we exclude untreated palms in nurseries, gardens of villas, or hotels, i.e., places where the owners have an economic interest in preserving their palms, and the economic resources to pay for the treatments. Figure 2 shows the evolution of the RPW distribution in the entire study area.

Considering the dead-infested palms among the treated ones (Figure 3), it is possible to notice a central area in which the concentration of the infested ones is higher compared to the remaining areas.

In fact, treatments were skipped in this area due to misunderstandings. This area is not under the care of municipal employees but of an external company. Our interviews revealed that there was often unclear communication between the two parties, and thus the omission of some treatments may have allowed the RPW to attack.

The chemicals employed in this period are pesticides usually used in open-field agriculture, while their authorizations for use in urban environments have changed over the years (Table 2).

### 3.2. Geo-Statistical Analysis

Comparing the hot spot map for treated palms with that of healthy palms, a very strong overlap can be observed (Figure 4). This means that where the palms were treated, there is a high concentration of healthy palms. Only small red areas (little hot spot clusters) are outside the areas covered by the municipality’s treatments (Figure 4).

By interpreting aerial and street-level photos (Google Earth, Google Maps, and Google Street) and field surveys, we found that the red dots in this hot spot cluster correspond to palm nurseries and the gardens of villas, farmhouses, or hotels.

This fact is best appreciated in the overlap map between treated healthy palms and untreated healthy palms (Figure 5). The index of the degree of overlap of 84.92% was generated based on the percentage of the number of squares superimposed between the two derived hot spot maps (20 × 20 m units). This percentage increases to 95% if we exclude the squares related to nurseries and the gardens of villas and hotels.

Finally, to support these spatial results with a statistical test, the Chi square test (X^2^) was calculated based on the RPW occurrence in “Treated” and “Untreated” category palms (starting values in Table 1). We performed two X^2^ tests, one considering all “Untreated” palms and another one subtracting those in nurseries or in the gardens of hotels, villas, or farmhouses. With 1 as the degree of freedom (contingency Table 2 × 2), the resulting values were X^2^ = 358.82 (*p*-value = 2.2^−16^) and X^2^ = 441.84 (*p*-value = 2.2^−16^), respectively. Therefore, the null hypothesis of independence was rejected: the occurrence of RPW depends on whether a given palm has been treated or not.

### 3.3. Computational Approach to Predict Imidacloprid- and Chlorpyrifos-Protein Interaction Networks

We built a chemical and biological interaction network to investigate the molecular pathways associated with the effects of IMI and CPF on five model organisms (*Apis mellifera*, *Gallus gallus*, *Danio rerio*, *Xenopus silurana*, and *Homo sapiens*). Using a confidence score >0.7 (a high confidence score, according to STITCH), we identified four main metabolic processes associated with proteins interacting with both chemicals (Table 3).

Data indicate that most of the identified proteins belong to the neurotransmitter-gated ion channel family. Neurotransmitter-gated ion channels constitute a very important group of transmembrane receptor-ion channel complexes in vertebrates and invertebrates. These receptors open transiently upon binding of specific ligands and govern synaptic neurotransmissions in the nervous system and neuromuscular transmissions [52]. Overall, our findings are not surprising, since they exactly match the insecticidal mode of action of the neonicotinoids [53]. Indeed, nicotinic acetylcholine receptors (nAChRs) are the main targets of neonicotinoid insecticides, including IMI. Structural analyses of nAChR have recently increased, revealing that the nitro group of IMI plays a pivotal role in the selective action of neonicotinoids on insect nAChR [54]. However, recent studies suggest possible adverse impacts of neonicotinoids on non-target organisms, including mammals [55,56]. IMI was found to activate nAChR signalling in neonatal rats [57] and cultured human neurons at concentrations that can be potentially reached by dietary or accidental exposure [58].

Along with the neurotransmitter-gated ion channels, high confidence scores were also found for genes involved in neurotransmitter catabolic processes, particularly acetylcholinesterase (AChE). The inhibition of AChE is a well-known mechanism triggered by CPF, and it is responsible for neurotoxic effects in insects. However, several studies suggest the potential of CPF to generate harmful effects in non-target species as well. For example, exposure to CPF has been associated with neurodevelopmental disorders in rodents and humans [59,60]. In amphibians, CPF was found to induce sublethal toxicity and indirect effects by impacting their ecosystem [61]. CPF is also known to have noncholinergic effects, such as inhibition of the oxidative metabolism of sex steroids [62]. This is in line with our analysis, which identified the cytochrome P450 (CYP) pathway as one of the key targets for CPF in each analyzed vertebrate model. Previous reports have shown that CPF induces reproductive disorders in mammals by affecting gametes [32,63] probably through oxidative stress mechanisms [64]. In the human model, we found that the cluster associated with metabolic pathways accounts for the largest number of CYPs and also includes genes encoding antioxidant enzymes such as CAT and enzymes involved in the choline pathway such as choline O-acetyltransferase (chat), butyrylcholinesterase (BCHE), and esterase (PON1). Similarly, two esterase proteins were found to be involved in biological processes associated with IMI/CPF exposure in *Apis millifera*. In conclusion, this computational approach was conducted to gain insight into the consequences of prolonged exposure to both IMI and CPF, widely used in the chemical control of *Rhynchophorus ferrugineus*. Our analysis demonstrates that for all organisms, both pesticides have a high toxic potential to induce neurotoxic effects and likely endocrine interference, especially in mammals (Figure 6).

These effects are closely related to their bioaccumulation capacity, which is evident for organochlorines, but they may also concern neonicotinoids. Indeed, IMI can persist for months in some soil conditions and may pose a risk to other organisms in the area. Thus, more research is needed for tracking these pesticides and their metabolites in the study area and for monitoring their long-term effects on non-target species, including humans.

## 4. Conclusions

Our monitoring program, carried out about 7 years after the previous one and about 13 years after the appearance of the RPW in the municipality of San Benedetto del Tronto, central Italy, showed how this beetle has markedly influenced the palm tree heritage in the study area, reducing it by more than half since 2007, when the RPW infestation began. In recent years, the San Benedetto del Tronto municipality, in concert with the Phytosanitary Service of the Marche Region, has actively worked to cope with the evolution of RPW infestations on palm trees, initially implementing large-scale integrated pest management and later, localized preventive treatments.

Our data show that the fight against the weevil is now concentrated in important municipal areas, namely the seaside avenues and parks, and at villas, hotels, farmhouses, and nurseries. In the rest of the territory, the large number of dead-infested palm trees observed in recent years clearly highlights the failure of large-scale integrated pest management compared to the detailed approach targeting seaside avenues and parks. Milosavljević et al. [2] identified inappropriate management planning, poor coordination between stakeholders, and public resistance to the implementation of treatments as potential causes of this failure, an assessment with which we substantially agree. The geospatial analysis approach used in this work demonstrates that well-implemented preventive chemical treatments are very effective in protecting the palms from RPW attack. However, this success is mitigated by concerns about public health and economic impacts related to the massive use of these substances. Indeed, the results of the system toxicology analysis reported here highlight potential hazards to non-target species due to repeated applications and long-term insecticide exposure.

In conclusion, the multidisciplinary approach adopted in the present study has proved to be very useful for evaluating the multiple aspects of the fight against RPW in an urban environment. In accordance with Sawyer and Casagrande [65], we believe that the long-term objective should be to elaborate new designs for urban environments that will minimize the negative interactions between people and pests, limiting the use of pesticides in urban areas. In this regard, it would be advisable to promote a diversified and therefore more resilient urban vegetal landscape with many different plant species, as this composition would more successfully weather possible new alien insect pest invasions.

## Figures and Tables

**Figure 1 insects-14-00502-f001:**
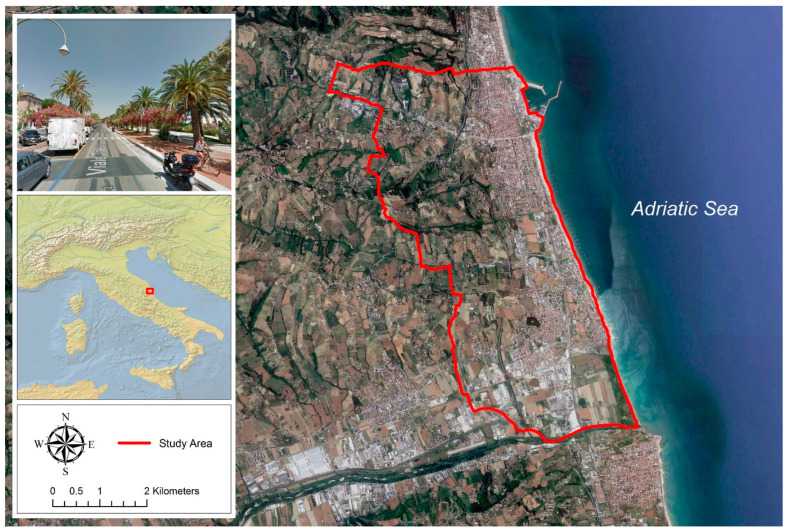
Location of the study area in the Marche region, central Italy (municipality of San Benedetto del Tronto).

**Figure 2 insects-14-00502-f002:**
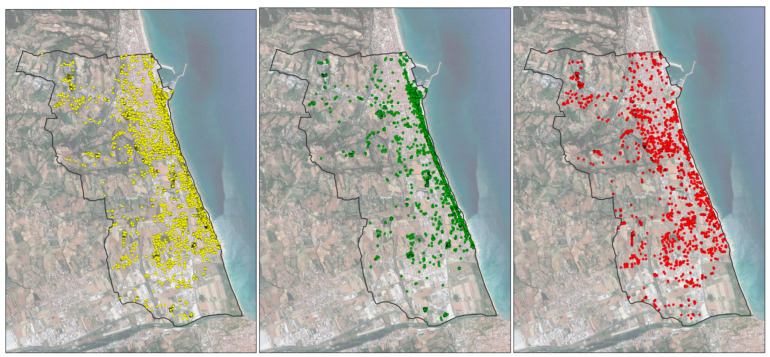
Evolution of the red weevil distribution in the study area (period 2013–2020); all palms in yellow, healthy palms in green, and dead-infested palms in red.

**Figure 3 insects-14-00502-f003:**
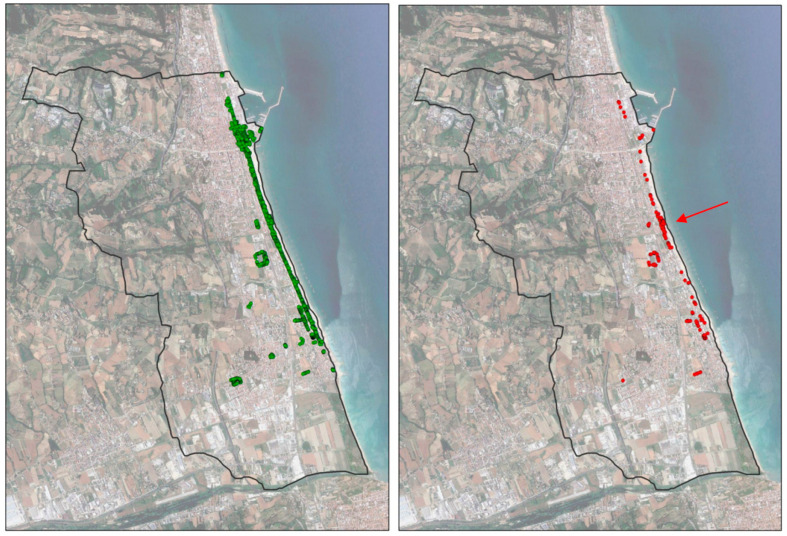
(**Left**), treated palms (green dots). (**Right**), dead-infested palm trees among the treated ones (red dots). The central area with the greatest concentration of dead-infested palm trees among the treated ones (red arrow) corresponds with the contact area between the direct management of the municipality to the north and that entrusted to an external company to the south.

**Figure 4 insects-14-00502-f004:**
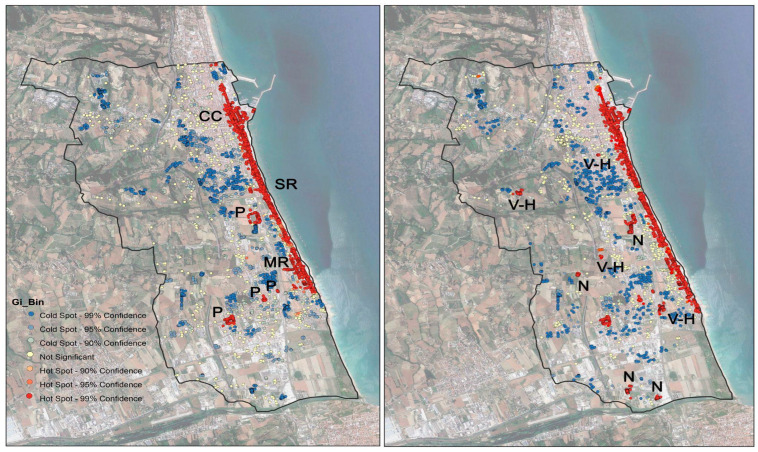
Hot spot analysis maps, with red indicating significant hot cluster groups of palms. (**Left**), chemically treated palms, (**Right**), healthy palms. SR: seaside road; CC: city center; MR: “Dei Mille” road, P: urban parks N: nursery; V-H: villas, hotels, or farmhouses.

**Figure 5 insects-14-00502-f005:**
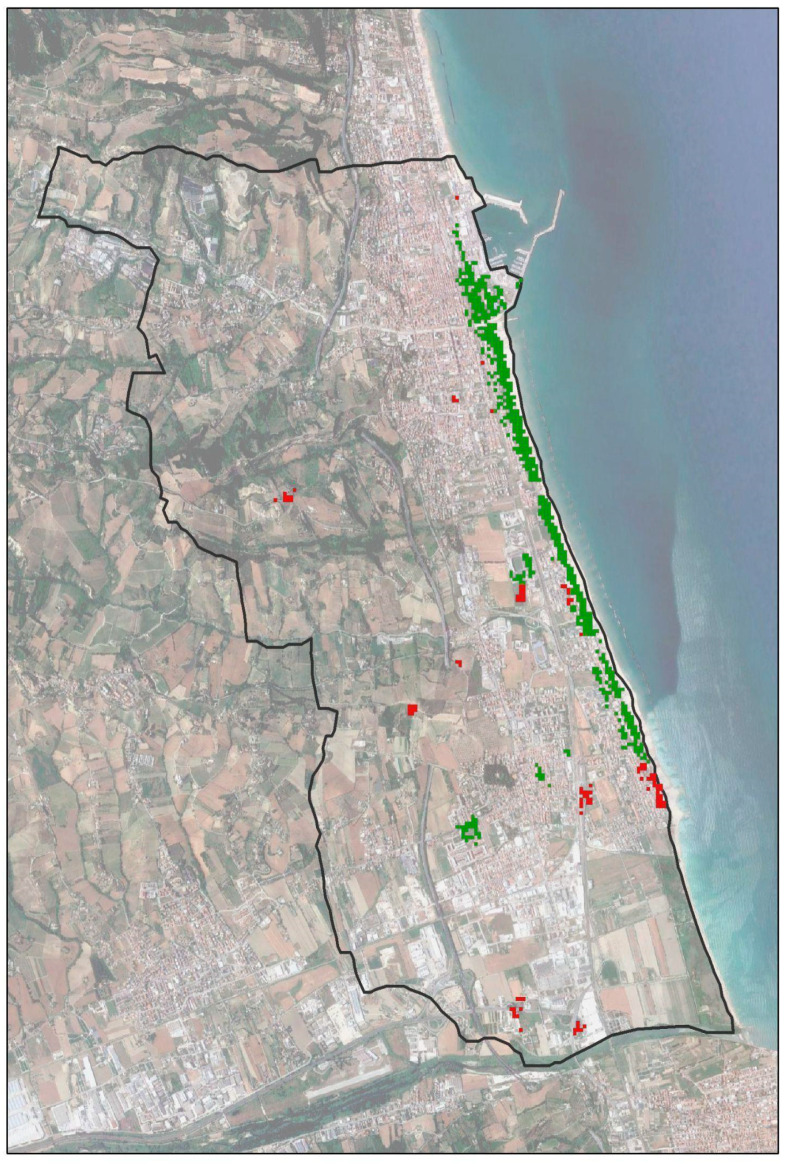
Assessment of the degree of overlap between hot spot analysis “healthy” vs. “treated” maps and analysis of anomalies. Green square: area with treated and healthy palms; red square: area with healthy palms not treated by the municipality. Square sampling: 20 × 20 m.

**Figure 6 insects-14-00502-f006:**
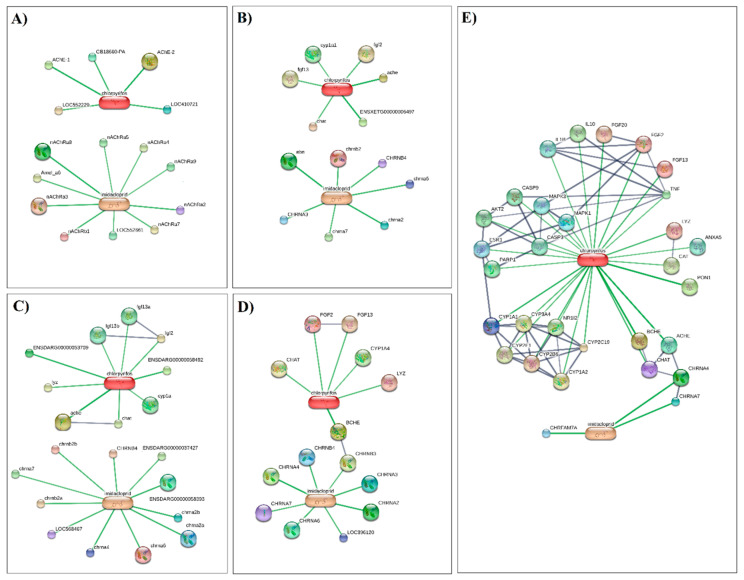
Interactions of imidacloprid and chlorpyrifos in (**A**) *Apis mellifera*, (**B**) *Xenopus silurana*, (**C**) *Danio rerio*, (**D**) *Gallus gallus*, (**E**) *Homo sapiens*. Chemicals are represented as pill-shaped nodes, while proteins are shown as spheres. Nodes that are associated with each other are linked by an edge with a confidence score higher than 0.7. All connections with uncharacterized proteins were removed to enhance visualization.

**Table 1 insects-14-00502-t001:** Classification of palms by health status and treatment status.

Category	Number
Total palms	7385
Healthy	3442
Dead-infested	3943 (of which 2456 have died since 2013)
Treated (since 2013)	1627
Healthy and treated	1460
Dead-infested and treated	167
Untreated	5758
Healthy and untreated	1982 (of which 1032 were in nurseries or villas/hotels)
Dead-infested and untreated	3776 (of which 272 were in nurseries or villas/hotels)

**Table 2 insects-14-00502-t002:** Authorized chemicals used in the study area against the RPW.

Chemicals	Year
Afidina quick (clorpirifos, deltamethrin)	2013–2014
Khoinor plus (imidacloprid, ciflutrin)	2015
Reldan (clorpirifos-metile)	2016–2019
Asset five (pyrethrins)	2020
Intercept granular (imidacloprid)	2013–2020

**Table 3 insects-14-00502-t003:** List of the main representative interactive pathways with imidacloprid and chlorpyrifos after enrichment analysis using STITCH v5.0.

Pathway ID	Pathway Description	#Protein	Matching Proteins in the Network
*Apis mellifera*			
IPR006201 IPR006202 IPR018000 IPR006029 AME04040	Neurotransmitter-gated ion channel family	8	nAChRa2, nAChRa3, nAChRa4, nAChRa5, nAChRa6, nAChRa7, nAChRa8, nAChRb1
GO:0042135 GO:0001507	Neurotransmitter catabolic process	3	AChE-1, AChE-2, LOC410721
AME00001	Metabolism	2	GB18660-PA, LOC552229
*Xenopus silurana*			
PF02931 PF02932	Neurotransmitter-gated ion channel family	8	Chrna3, chrnb4, ENSXETG00000018003, chrna2, chrna6, chrna7, chrnb2, ebn
GO:0006581	Neurotransmitter catabolic process	1	AChE
XTR00199 XTR00001	Metabolism	2	CYP1A1, chat
XTR04052	Signalling and cellular processes	2	fgf2, fgf13
*Danio rerio*			
PF02931 PF02932	Neurotransmitter-gated ion channel family	11	Chrnab4, ENSDARG00000037427, ENSDARG00000058393, LOC568467, chrna2a, chrna2b, chrna4, chrna6, chrna7, chrnb2a, chrnb2b
GO:0042135	Neurotransmitter catabolic process	3	AChE, ENSDARG00000053709, ENSDARG00000058492
DRE00199 DRE00001	Metabolism	2	CYP1A1, chat
DRE04052 DRE03037	Signalling and cellular processes	3	fgf2, fgf13a, fgf13b, lyz
*Gallus gallus*			
PF02931 PF02932 GGA04040	Neurotransmitter-gated ion channel family	8	Chrna6, chrna7, chrnb3, LOC396120, chrna3, chrna4, chrnb4, chrna2
GGA04147	Signalling and cellular processes	3	LYZ, fgf2, fgf13
GGA01000 GGA00199	Metabolism	3	BCHE, CYP1A4, chat
*Homo sapiens*			
GO:0005230	Neurotransmitter-gated ion channel family	3	Chrna4, Chrna7, Chrfam7A
GO:0042136	Neurotransmitter catabolic process	1	AChE
GO.0017144 GO:0006805 GO:0006629 GO:0002682 GO:0044255 GO:1901575	Metabolism	11	BCHE, CYP1A1, CYP1A2, CYP2B6, CYP2C19, CYP2E1, CYP3A4, NR1I2, chat, CAT, PON1
GO:0032870 GO:0008286 GO:0009967 GO:0060397 GO:0060396 HSA04010 IPR008996	Signalling and cellular processes	16	AKT2, CASP9, CASP3, ESR1, FGF2, FGF20, IL10, FGF13, TNF, MAPK1, MAPK3, NR1I2, PARP1, LYZ, ANXA5, IL1B

## Data Availability

The data presented in this study are available on request from the corresponding author. The data are not publicly available as they are partly owned by local governments.

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
