# Peer review of "Multiple Aspects of the Fight against the Red Palm Weevil in an Urban Area: Study Case, San Benedetto del Tronto (Central Italy)"

_insects, 2023, doi:10.3390/insects14060502_

Round 1

Reviewer 1 Report

This article by Luca Bracchetti and colleagues is well motivated, the structure is somewhat appropriate, and the manuscript is not missing any key details. The methods used are appropriate for the objectives of the work and, the resulting figures are sufficient, informative, and of good quality helping to follow the reasoning throughout the manuscript. Comments on future research was nicely done and will be useful to others.

My primary concern is about the language. I had difficulty understanding the English at times, which affected my ability to understand whether the authors have entirely interpreted their results appropriately. There are quite a lot of errors (awkward phrasing, poor vocabulary, syntax and grammar errors, etc).

My other concern is about the formatting. Species’ names and test statistics should be italicized throughout the manuscript. References should be numbered.

I suggest minor revision that will consider what I had in mind.

I had difficulty understanding the English at times, which affected my ability to understand whether the authors have entirely interpreted their results appropriately. There are quite a lot of errors (awkward phrasing, poor vocabulary, syntax and grammar errors, etc).

Author Response

A native English speaker has reviewed the manuscript.

The formatting of some words has been changed.

The references have been numbered.

Reviewer 2 Report

It is a good research paper, though, it does not follow the journal's standards and it is badly written. The use of English language is poor (I think in many cases are typos, like Figure 2 No treated instead of Not treated), scientific names are not in italics (e.g., Line 14, Line 122), or incorrect use of italics (Line 53-54 families must not be in italics), incorrect usage of metric abbreviations (Line 149 mt for meters), and of course the references. Please reconsider all these before resubmitting

A lot of English language mistakes can be found, making it difficult to read (e.g., Line 58you have to use plural methods  instead of method, Lines 94-96 makes no sense. I stronglly suggest to give the manuscript to a fluen English speaker o duble check all the mistakes.

Author Response

A native English speaker has reviewed the manuscript.

The formatting of some words and the metric abbreviations has been changed.

The references have been numbered.

Reviewer 3 Report

This is an interesting article that aims at investigating the evolution of the red palm weevil and the efficacy of the chemical products used to control the damage provoke by this exotic pest. They also used a model to evaluate the potential impact of the insecticides on other organisms and human health. I think the methodology used is innovative and robust and this article will make a great contribution to the fight against this weevil on urban environments. However, I think the paper needs some work before it can be published. I offer the following comments for consideration:

·       Lines 26-27: I think it is not necessary to say in the abstract that this paper is an upgrade of a previous paper. You should focus on your results and on explaining how this study may help in the fight against this weevil.

·       Introduction: You should include a brief description of the life cycle of the insect and the kind of damage it provokes to help the potential readers that are not familiar with this weevil to understand the importance of you work.

·       Lines 46-47: Is it really necessary to cite all of these EPPO reports? Is there not any reference that can replace all these EPPO reports? Milosavjevic et al. (2019) or Rochat et al. (2006) may be a good option.

·       Line 69: If you are going to use the acronym RPW to refer to this insect, you should be consistent and use it throughout the manuscript.

·       Lines 134-141: I think you should expand this section to give more details about the hotspot analysis so the potential reader that is not familiar with this technique can understand what you did here. For example, you should provide the formula for the Gi* statistics. Furthermore, this statistic is sensible to distance threshold (d). So, how did you determine the d value? Did you use the default value? How would affect your results a change in this value? Also, you mention spatial clusters with high and low values, do these make reference to hotspots and coldspots? What are the z-scores values that render a cluster aggregation statistically significant? Etc.

·       Lines 158-159: not clear what you mean by this number. Please clarify.

·       Line 176: What does SMILES stand for?

·       Table 1: 3943 (on which..)..Do you mean of which?

·       Line 248: …mining…Do you mean meaning?

Lines 440 and 442: It would be useful to add an english translation of these titles to have an idea of what are these papers abou

Author Response

Our responses to comments are in bold

  • Lines 26-27: I think it is not necessary to say in the abstract that this paper is an upgrade of a previous paper. You should focus on your results and on explaining how this study may help in the fight against this weevil.

The abstract has been modified.

  • Introduction: You should include a brief description of the life cycle of the insect and the kind of damage it provokes to help the potential readers that are not familiar with this weevil to understand the importance of you work.

We have included both a brief description of the life cycle of the insect and the kind of damage it provokes.

  • Lines 46-47: Is it really necessary to cite all of these EPPO reports? Is there not any reference that can replace all these EPPO reports? or Rochat et al. (2006) may be a good option.

We have changed the references as suggested.

  • Line 69: If you are going to use the acronym RPW to refer to this insect, you should be consistent and use it throughout the manuscript.

Appropriate changes made to the text.

  • Lines 134-141: I think you should expand this section to give more details about the hotspot analysis so the potential reader that is not familiar with this technique can understand what you did here. For example, you should provide the formula for the Gi* statistics. Furthermore, this statistic is sensible to distance threshold (d). So, how did you determine the d value? Did you use the default value? How would affect your results a change in this value? Also, you mention spatial clusters with high and low values, do these make reference to hotspots and coldspots? What are the z-scores values that render a cluster aggregation statistically significant? Etc.

We have expanded this section giving more details about both the hot spot analysis setting and interpretation of its values; we have also entered the formula as required.

  •     Lines 158-159: not clear what you mean by this number. Please clarify.

It was a datum of the analyzes that should not have been reported here; part deleted.

  • Line 176: What does SMILES stand for?

SMILES is acronym for Simplified Molecular Input Line Entry System; we have added this explanation.

  • Table 1: 3943 (on which..)..Do you mean of which?

Appropriate changes made to table 1.

  • Line 248: …mining…Do you mean meaning?

Appropriate changes made to the text.

Lines 440 and 442: It would be useful to add an english translation of these titles to have an idea of what are these papers about

Translation included in these reference.

Round 2

Reviewer 1 Report

Authors have done a nice job addressing all of my original comments and those of other reviewers. Thank you.

Author Response

We are happy for this comment.